# Remarkable Nonlinear Properties of a Novel Quinolidone Derivative: Joint Synthesis and Molecular Modeling

**DOI:** 10.3390/molecules27082379

**Published:** 2022-04-07

**Authors:** Clodoaldo Valverde, Rafael S. Vinhal, Luiz F. N. Naves, Jean M. F. Custódio, Basílio Baseia, Heibbe Cristhian B. de Oliveira, Caridad N. Perez, Hamilton B. Napolitano, Francisco A. P. Osório

**Affiliations:** 1Campus de Ciências Exatas e Tecnológicas, Universidade Estadual de Goiás, Anápolis 75001-970, GO, Brazil; luiz.fn@gmail.com (L.F.N.N.); hamilton@ueg.br (H.B.N.); 2Universidade Paulista — UNIP, Goiânia 74845-090, GO, Brazil; 3Instituto de Física, Universidade Federal de Goiás, Goiânia 74690-900, GO, Brazil; siqueirarsv@gmail.com (R.S.V.); basiliobaseia@yahoo.com.br (B.B.); fosorio76@gmail.com (F.A.P.O.); 4University of Notre Dame, Notre Dame, IN 46656, USA; jeanmfcustodio@gmail.com; 5Departamento de Física, Universidade Federal da Paraíba, João Pessoa 58051-970, PB, Brazil; 6Instituto de Química, Universidade Federal de Goiás, Goiânia 74690-900, GO, Brazil; heibbe@ufg.br (H.C.B.d.O.); caridad@ggmail.com (C.N.P.); 7Escola Politécnica, Pontifícia Universidade Católica de Goiás, Goiânia 74605-100, GO, Brazil

**Keywords:** third-order, Hirshfeld, X-ray diffraction

## Abstract

A novel 4(1H) quinolinone derivative (QBCP) was synthesized and characterized with single crystal X-ray diffraction. Hirshfeld surfaces (HS) analyses were employed as a complementary tool to evaluate the crystal intermolecular interactions. The molecular global reactivity parameters of QBCP were studied using HOMO and LUMO energies. In addition, the molecular electrostatic potential (MEP) and the UV-Vis absorption and emission spectra were obtained and analyzed. The supermolecule (SM) approach was employed to build a bulk with 474,552 atoms to simulate the crystalline environment polarization effect on the asymmetric unit of the compound. The nonlinear optical properties were investigated using the density functional method (DFT/CAM-B3LYP) with the Pople’s 6-311++G(d,p) basis set. The quantum DFT results of the linear polarizability, the average second-order hyperpolarizability and the third-order nonlinear susceptibility values were computed and analyzed. The results showed that the organic compound (QBCP) has great potential for application as a third-order nonlinear optical material.

## 1. Introduction

The 4(1H)-quinolinones make up a class of compounds that occur naturally in nature and are found in many plants, as well as being synthesized artificially. Among all heterocyclics, quinolinone derivatives form a group with privileged structures that exhibit various biological activities such as antibiotic, anticancerous, anti-inflammatory, antidiabetic and antipsychotic actions. This wide spectrum of biological activities has attracted a great deal of attention in recent years in the field of chemistry applied to medicine [1,2].

Interest in nonlinear organic materials with large third-order macroscopic nonlinear susceptibility values has increased in recent years, due to their moderately large nonlinearities, great structural flexibility, and very fast response time [3,4,5,6,7]. In the last two decades, a general rule that serves for the selection and choice of organic materials with good third-order nonlinear optical properties has yet to be established. Common projects to design potentially good optical materials are those that involve conjugated structures of electron donors and/or electron acceptors [6,7,8,9,10], and organic structures with good charge fluidity, such as diradical and zwitterionic compounds. Recently these structures have been suggested as the key structures for this field of research. In this context, the nonlinear optical properties of quinolinone derivative structures have appeared with some frequency in the literature due to the electron deficiency characteristics of these structures [8,9,10] and to their great potential application in the field of Organic Light Emitting Diodes (OLEDs) [11,12,13,14].

In the present work, the synthesis and crystallographic characterization of a new quinolinone derivative, namely, (E)-3-(4-Bromobenzylidene)-2-(4-chlorophenyl)-2,3-dihydro-1-(phenylsulfonyl)-quinolin-4(1H)-one (QBCP), with the formula C_28_H_19_BrClNO_3_S, are described in detail. The Hirshfeld surfaces (HS) and shape index analysis were used to evaluate the intermolecular interactions and supramolecular arrangements in the crystalline state. The supermolecule approach (SM) was employed to simulate the crystalline environment polarization on the asymmetric unit of a QBCP molecule. The linear and nonlinear optical (NLO) parameters, such as the dipole moment, linear polarizability, linear refraction index, second order hyperpolarizability and third-order nonlinear susceptibility were calculated at the CAM-B3LYP/6-311++G(d,p) chemistry model. The static and dynamic cases were considered and the results for the third-order nonlinear susceptibility were compared with the experimental results obtained for other organic materials. Furthermore, complementary information about the molecular properties, such as the frontier molecular orbitals, the molecular electrostatic potential (MEP) map and the UV-vis absorption and emission spectra were theoretically obtained and analyzed.

## 2. Results

### 2.1. Synthesis and Crystallization

The synthesis and chemical characterization of (E)-3-(4-Bromobenzylidene)-2-(4-chlorophenyl)-2,3-dihydro-1-(phenylsulfonyl)-quinolin-4(1H)-one was described by [15]. Chalcone (E)-3-(4-Chlorophenyl)-1-(2-(phenylsulfonylamine)phenyl)prop-2-en-1-one (1.0 mmol) and 4-bromobenzaldehyde (2.0 mmol) were dissolved in 15 mL of ethanol with 56.1 mg of dissolved potassium hydroxide. The reaction was performed at 25 °C over 48 h. The reaction solution was then filtered, and the precipitate rinsed with 15 mL of ethanol. After that, the precipitate was dissolved in 10 mL of dichloromethane and extracted with water. The organic phase was slowly evaporated, yielding the product crystals. The reaction yield and the purity were of 76.1% and 98.2%, respectively. The recrystallization to obtain more pure crystals was achieved by dissolution in dichloromethane and exposure to diethyl ether vapors.

### 2.2. Crystallographic Characterization

X-ray diffraction data collection was performed by a Bruker APEX-II CCD diffractometer using Mo*K*α radiation with wavelength of 0.71073 Å. The crystallographic structure was solved by direct methods using the OLEX2 routine [16] in the Olex2 1.3 [17] software, where refinement was executed by SHELXL 2016/6 [18]. Hydrogen atoms from phenyl and 4-bromophenyl rings were refined by an idealized distances and angles restrain encouraged by the presence of disorder in these regions, while the other hydrogens were refined freely. Crystal data and the refinement parameters may be accessed in the Cambridge Crystallographic Data Centre (CCDC) [19] free of charge via the web address ccdc.cam.ac.uk (accessed on 4 April 2022) under the deposit code 2150000. Mercury [20] and ORTEP [21] were utilized to visualize and analyze the crystal structure, as well as to map contacts and understand the interactions. Structure validation was performed by PLATON [22] and a Crystal Explorer17 was used to calculate the Hirshfeld surface and its properties [23].

The Hirshfeld surface $\omega \left(r\right)$ is defined by the region where the electronic density contribution of the molecule to the crystal ($\rho_{promolecule}$) surpasses the electronic density contribution of other molecules on the crystal ($\rho_{procrystal}$) [24]. Some properties of this surface are used to investigate intermolecular interactions in crystals [25], such as the distance between a point on the surface and the nearest nucleus outside the surface ($d_{e}$) or the nearest nucleus inside the surface ($d_{i}$). The frequency of the $d_{e}$ and $d_{i}$ in the Hirshfeld surface was mapped with the 2D fingerprint plot feature [26,27], which gives a unique diagram of intermolecular interactions in the crystal, filterable by element involved in the interaction.

### 2.3. Theoretical Analysis of the Molecular Properties

The frontier molecular orbital (FMO) energies, the highest occupied molecular orbital (HOMO) and the lowest occupied molecular orbital (LUMO), are important parameters for the understanding of the molecular chemical reaction. HOMO behaves as an electron donor and LUMO as an electron acceptor. FMO energies can be used to calculate the global parameters of chemical reactivity. The molecular electrostatic potential (MEP) and the CHELPG partial charges were obtained and analyzed for both isolated and embedded molecules. Moreover, the UV-Vis absorption and emission spectra of the QBCP molecule calculated at the time-dependent TD-DFT/CAM-B3LYP/6-311++G(d,p) level of theory in a vacuum are presented. Both ground (S_0_) and first excited (S_1_) states were fully optimized and characterized by frequencies calculations.

### 2.4. The Crystalline Environment Simulation

The crystalline environment polarization effect on the asymmetric unit of a QBCP molecule was considered via the supermolecule (SM) method, where the surrounding molecules are considered as point charges [28,29,30,31,32,33,34,35,36,37,38,39,40,41,42,43,44,45]. The SM method is an interactive process that starts with the adjustment of the molecular electrostatic potential through the ChelpG scheme considering the electric charge distributions in a vacuum and the partial charges of the asymmetric unit are calculated. After that, all the charges of the surrounding atoms of the asymmetric unit are changed by the partial charges calculated in the previous step. Then the dipole moment and the new set of the molecular partial charges are calculated. The process of the substitution of the partial charges continues until the magnitude of the dipole moment converges. In the present work, a bulk with a 13 × 13 × 13-unit cell was built with four asymmetric units per unit cell and 54 atoms in each asymmetric unit, totaling 474,552 atoms. It is worth noting that the QBCP geometry used in the SM method was retrieved from the crystallographic file and possible disorder was not taken into account.

Appendix A (Support Information) shows the unit cell of the QBCP structure with the four asymmetric units, whose design is indicated by A, B, C and D. To check whether the choice of asymmetric unit influenced the results, the crystalline environment was built for each one of these asymmetric units (A, B, C and D) to simulate the crystal. For each one of the four embedded molecules, Appendix A in the Support Information file shows the *μ*—steps of convergence. The electric properties of the simulated crystals shown below were independent of the asymmetric unit choice. Figure 1 shows the ORTEP representation of the QBCP molecule (a) and the convergence of the total dipole moment (*μ*) as a function of the iterative steps (b). Note that the convergence of the dipole moment was reached after five steps at 5.05 D and that no difference due to the chosen asymmetric unit was observed.

### 2.5. Linear and Nonlinear Optical Parameters

To obtain the linear parameters of the QBCP crystal, the numerical evaluation employed the following definitions. The total dipole moment is given by:(1)$$\mu =\sqrt{\mu_{x}^{2}+\mu_{y}^{2}+\mu_{z}^{2}}\;,$$
the average linear polarizability ($\langle \alpha \left(-\omega ;\omega \right)\rangle$) and the linear refractive index ($n\left(\omega \right)$), were calculated by the following relationship:(2)$$\langle \alpha \left(-\omega ;\omega \right)\rangle =\frac{1}{3}\overset{}{\underset{i=x,y,z}{\sum }}\alpha_{ii}\left(-\omega ;\omega \right),\;$$
(3)$$\frac{n{\left(\omega \right)}^{2}-1}{n{\left(\omega \right)}^{2}+2}=\frac{4\pi N}{3\;V}\langle \alpha \left(-\omega ;\omega \right)\rangle ,$$
where Equation (6) is the Clausius–Mossotti relation, N and Vare the number of asymmetric units and the volume of the unit cell and ω is the applied electric field frequency.

The average second hyperpolarizability ($\langle \gamma \left(-\omega ;\omega ,0,0\right)\rangle$) is defined by:(4)$$\begin{array}{ll} \gamma \left(-\omega ;\omega ,0,0\right) & =\frac{1}{5}\left(\gamma_{xxxx}+\gamma_{yyyy}+\gamma_{zzzz}\right)\\ & +\frac{1}{15}\left[\gamma_{xxyy}+\gamma_{yyxx}+\gamma_{xxzz}+\gamma_{zzxx}+\gamma_{yyzz}+\gamma_{zzyy}+4\left(\gamma_{yxyx}+\gamma_{zxzx}+\gamma_{zyzy}\right)\right]. \end{array}$$

From Equation (5), the average IDRI (intensity-dependent refractive index) second hyperpolarizability was obtained from the approximation:(5)$$\langle \gamma \left(-\omega ;\omega ,\omega ,-\omega \right)\rangle ≅2\langle \gamma \left(-\omega ;\omega ,0,0\right)\rangle -\langle \gamma \left(0;0,0,0\right)\rangle .$$

Finally, the dynamic third-order nonlinear susceptibility is given by:(6)$$\chi^{\left(3\right)}\left(-\omega ;\omega ,\omega ,-\omega \right)={\left(\frac{\left(n^{2}+2\right)}{3}\right)}^{4}\frac{N\langle \;\gamma \left(-\omega ;\omega ,\omega ,-\omega \right)\rangle }{V}.$$

All the computational calculations including the optical properties were performed at the DFT/CAM-B3LYP/6-311++G(d,p) level of theory as implemented at the Gaussian-16 program package [46]. The CAM-B3LYP/6-311++G(d,p) level of theory has been shown to be accurate at performing calculations of NLO and photophysical properties [38,47].

## 3. Discussion

### 3.1. Solid State Description

The QBCP molecule is a 4(1H)-quinolinone (ring A) derivative with substituents (E)-4-bromobenzylidene (ring B), 4-chlorophenyl (ring C) and phenylsulfonyl (ring D), as shown in Figure 2. The solid structure of QBCP crystallizes in the centrosymmetric monoclinic space group *P*2_1_/*n* with one molecule in an asymmetric unit and four asymmetric units per unit cell. The main crystal data and refinement parameters can be found in Table 1.

Appendix A presents the ORTEP representation showing the presence of disorder in rings B and D, denoted by atoms labeled with and without the suffix A. For the purposes of clarity, only the atoms labeled without a suffix will be displayed and only the contacts that are present in both cases will be discussed. The essential dihedral angles to describe the QBCP conformation are present in Table 2. As the structure is centrosymmetric, the presence of a chiral center located at C10 causes this compound to consist of a racemate and every interaction is replied by its enantiomer. Therefore, there is no need to describe the interactions of each enantiomer separately.

Since QBCP presents no acid hydrogens, its stronger intermolecular interactions consist of non-classical C—H···O contacts. The C6—H6···O3 and C25—H25···O2 contacts create different paths of interactions but in both the crystallographic directions (010), as seen in Figure 3a and Figure 3b, respectively. Together these contacts build a layer of molecules that lie in the crystallographic plane (001). In addition, the C18—H18···O1 interaction forms a dimer between a pair of enantiomers (Figure 3c) and connects the layers of the packing, stacking them in the (001) direction. Table 3 shows the geometric parameters of these interactions.

Aromatic rings have weaker but relevant intermolecular interactions . The center of each of them was calculated and named Cg1, Cg2 and Cg3 for rings B, C and D, respectively, to measure the atom···π contacts distances (Table 4). Figure 4a shows the Br···π interaction involving the ring C that forms a path of interactions in the (100) directions, where the Cl···π interaction with ring D (Figure 4b) and the O···π interaction with ring B (Figure 4c) contributes to the stabilization of the crystal packing. These interactions were also identified in the analysis of the Hirshfeld surface properties (Appendix A). The $d_{norm}$ property is represented on red spots between atoms (Appendix A) while the fingerprints show the frequency of the type of interactions in the structure. The shape index is shown by the red concave regions, a characteristic of the aromatic ring involved in atom···π interactions (Appendix A). The intermolecular interaction frequency is mapped in a fingerprint 2D plot (Appendix A). Due to the high presence of hydrogens on the exterior of the molecule, most of the interactions occurred between H and C atoms.

### 3.2. Molecular Modeling Analysis

The molecular charge distribution can be seen in Figure 5 where the molecular electrostatic potential (MEP) map is presented for both isolated (a) and embedded (b) molecules of QBCP. The blue color represents the positive electrostatic potential regions while the red color represents the negative electrostatic potential regions, located mainly in the vicinity of the oxygen atom indicating the maximum of the electronic density.

Table 5 shows the charge of the selected portions (groups) of the isolated and embedded molecules of QBCP. The groups 1,2 and 3 of the embedded molecules have the positive charges enhanced as compared with the isolated molecule results. For the benzene ring (group 2), the charge increased the respective value in the isolated molecule by 5.75 times. The charge of group 4 was practically unaltered; however, group 5 (SO_2_) presented a strong reduction in the absolute value of the charge, accomplished with a charge signal change (from positive to negative).

Figure 6 shows the plots of the HOMO and LUMO orbitals for the isolated and embedded molecules of QBCP. HOMO orbitals in both cases were concentrated over the (*E*)-4-bromobenzylidene portion, while the LUMO orbital was concentrated over the isolated molecule, except over the 4-chlorophenyl portion and the benzene ring. For embedded molecules, the difference was that the HOMO did not include the entire phenylsulfonyl portion. Table 6 shows the respective energy values of HOMO ($\varepsilon_{HOMO})$ and LUMO ($\varepsilon_{LUMO})$ orbitals and the global reactivity descriptors. The gap energy is defined as the difference between the HOMO and LUMO energies, as shown in Figure 6. The gap energies (E) for the isolated and embedded molecules were 6.53 eV and 6.43 eV, the difference being due to the small crystalline polarization effect, of 0.1 eV.

The global chemical descriptors can be obtained from the HOMO and LUMO energies. These quantities shown in Table 6 are frequently used as complementary tools in the description of the thermodynamic aspects of the chemical reactivity in connection with minimum polarizability and maximum hardness principles. In Table 6, besides the gap energies, we calculated the ionization energy ($I_{E}$), electron affinity energy ($A_{E})$, global hardness (η), chemical potential (μ), global softness (σ) and the global electrophilicity (ϖ). The ϖ-value is related to the charge transfer process and the ability of donating ($\varpi^{-})\;$or accepting ($\varpi^{+})\;$an electron of a molecule can be obtained from the descriptors:(7)$$\varpi^{-}=\frac{{\left(3I_{E}+A_{E}\right)}^{2}}{16\left(I_{E}-A_{E}\right)},$$
and
(8)$$\varpi^{+}=\frac{{\left(I_{E}+3A_{E}\right)}^{2}}{16\left(I_{E}-A_{E}\right)}.$$

The ionization energy value of a QBCP-isolated molecule is higher than the electron affinity value, which characterizes a higher electron donating capacity ($\varpi^{-}$) compared to the capacity of the electron accept (ϖ^+^). The crystalline environment polarization effect decreases the electron donating capacity ($\varpi^{-}$) of the QBCP-embedded molecule compared to the electron acceptor capacity (ϖ^+^). The results shown in Table 6 indicate that the effects of the crystalline environment polarization on the investigated structure decreased its global descriptors, keeping the global softness constant, indicating that the studied structure was chemically hard with a greater kinetic stability and electron donating capacity.

Figure 7 shows the normalized absorption and emission spectra of a QBCP-isolated molecule calculated at the time-dependent TD-DFT/CAM-B3LYP/6-311++G(d,p) level of theory. In this figure, the absorption spectrum showed two prominent peaks related to the stronger transition at 235.24 nm with an oscillator strength of 0.138 and at 294.18 nm and 286.68 nm with oscillator strengths of 0.221 and 0.315, respectively. The maximum peak of the emission spectrum was red-shifted by 31.8 nm in relation to the maximum peak of absorption.

### 3.3. Optical Parameter Descriptions

Table 7 shows the CAM-B3LYP/6-3111++G(d,p) results for the static and dynamic linear refraction index ($n\left(\omega \right)$), the linear polarizability ($\alpha \left(-\omega ;\omega \right)$), the average second hyperpolarizabilities $\langle \gamma \left(-\omega ;\omega ,0,0\right)\rangle$ and 〈γ(−ω;ω,ω,−ω〉 corresponding to the Kerr effect and to the IDRI effect, respectively; and the nonlinear third-order susceptibility ($\chi^{3}\left(-\omega ;\omega ,\omega ,-\omega \right)$) for the QBCP-embedded molecules. The electric field frequencies considered in the calculations were ω = 0.0426 *a.u.* (1064 nm) and ω = 0.086 *a.u.* (532 nm), these frequencies being commonly used in a nonlinear optical experimental arrangement. In Table 7 it may be observed that the values of the electric parameter increased with the increasing frequency. The dynamic linear optical parameter values, refractive index and linear polarizability present a percentage increase from 1.2% to 4.3% (1064 nm) and from 1.7% to 8.1% (532 nm), respectively when compared with the static values. The average second hyperpolarizabilities, $\gamma \left(-\omega ;\omega ,0,0\right)$*,* and $\gamma \left(-\omega ;\omega ,\omega ,-\omega \right)$, increased 89.5% (13.7%) and 179.1% (27.4%),respectively, in relation to the static values for λ = 532 nm (λ = 1064 nm). The value of the third-order nonlinear susceptibility, $\chi^{3}\left(-\omega ;\omega ,\omega ,-\omega \right)$ at λ = 532 nm is $2.79\times 10^{-20}m^{2}/V^{2}$, a significant value when compared with experimental values obtained for other organic structure (see Table 8), indicating very good NLO properties for QBCP.


## 4. Conclusions

In the present study we reported the synthesis of a new quinolinone derivative. The structure was characterized by single crystal X-ray diffraction (SCXRD) and by theoretical tools, such as the Hirshfeld surfaces (HS), 2D fingerprints and the shape index analysis. A bulk with a 13 × 13 × 13-unit cell and 474,552 atoms was built using an iterative electrostatic polarization scheme to simulate the crystalline environment polarization effect on an asymmetric unit of QBCP. Then the molecular properties of isolated and embedded molecules were obtained at the CAM-B3LYP/6-311++G(d,p) chemistry model. The global chemical descriptors were calculated from the HOMO and LUMO energies and the values for the QBCP-isolated and embedded molecules showed that they are hard molecules with a great capacity to donate electrons. The UV-Vis absorption and emission spectra of QBCP were studied and the Stokes shift calculated. The linear and NLO parameters, such as the dipole moment, linear polarizability, linear refraction index, second order hyperpolarizability and third-order nonlinear susceptibility were calculated at a static regime and for two electric field frequencies. The values of the NLO parameter increased with the increase in the field frequency and the values obtained were significant, indicating very good NLO properties exhibited by the QBCP. Particularly, the third-order nonlinear susceptibility value at 532 nm, $2.79\times 10^{-20}m^{2}/V^{2}$, was a value greater than that obtained for other organic materials; therefore, QBCP has the potential to be considered as a good candidate for applications using NLO material.

## Figures and Tables

**Figure 1 molecules-27-02379-f001:**
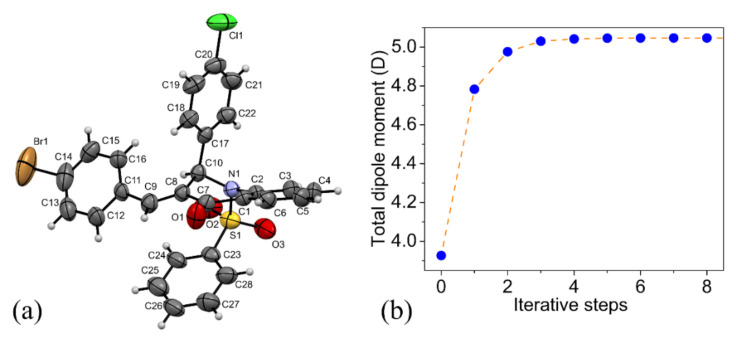
(**a**) The ORTEP representation of the asymmetric unit; (**b**) The total dipole moment as a function of the iterative steps for the QBCP simulated crystal.

**Figure 2 molecules-27-02379-f002:**
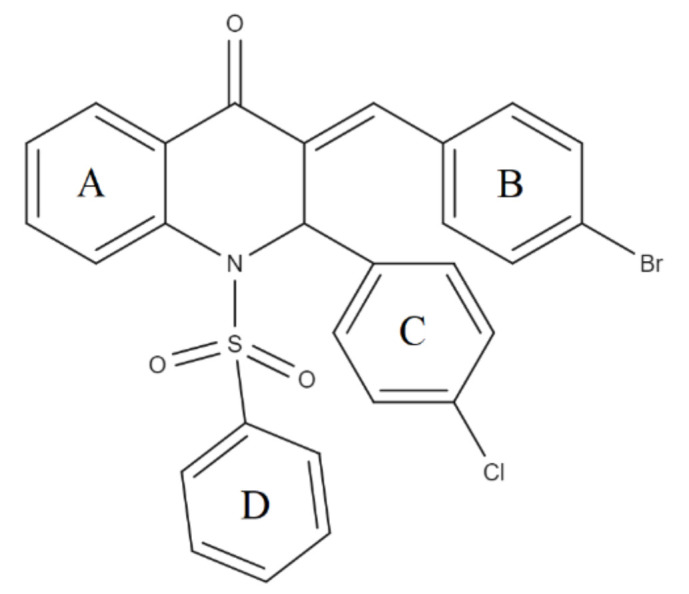
Structural formula of QBCP.

**Figure 3 molecules-27-02379-f003:**
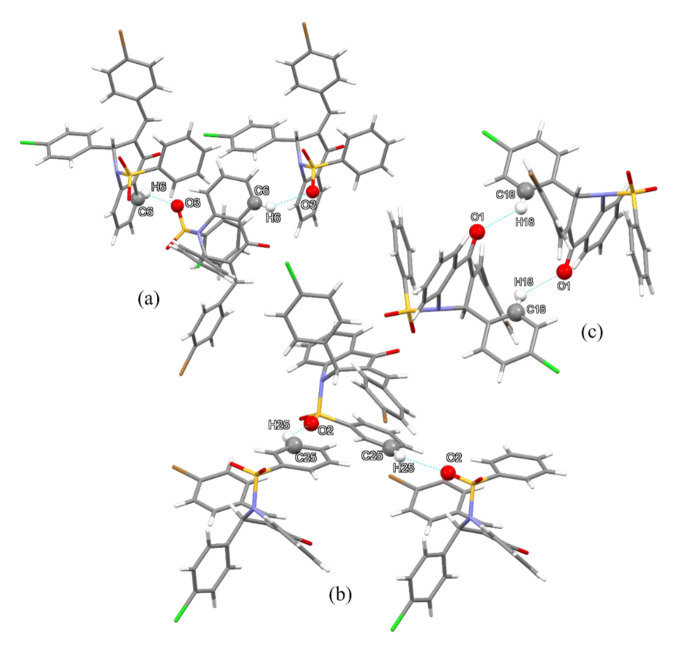
C6—H6···O3 (**a**) and C25—H25···O2 (**b**) interactions maintain a layered patter stacked by C18—H18···O1 (**c**) interactions.

**Figure 4 molecules-27-02379-f004:**
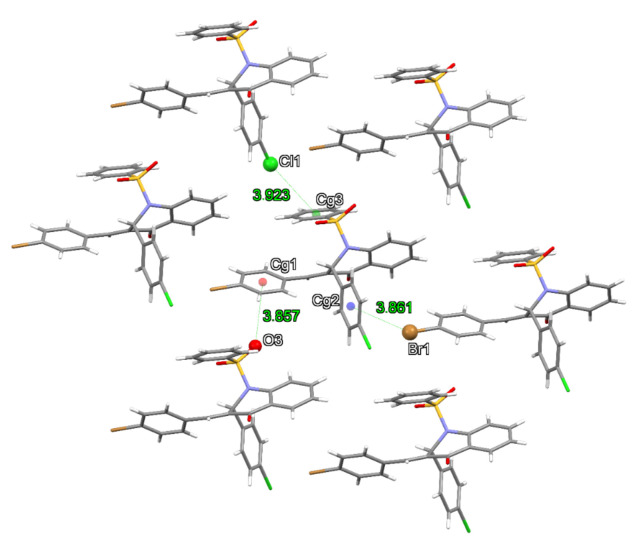
Contacts involving aromatic rings in the QBCP structure.

**Figure 5 molecules-27-02379-f005:**
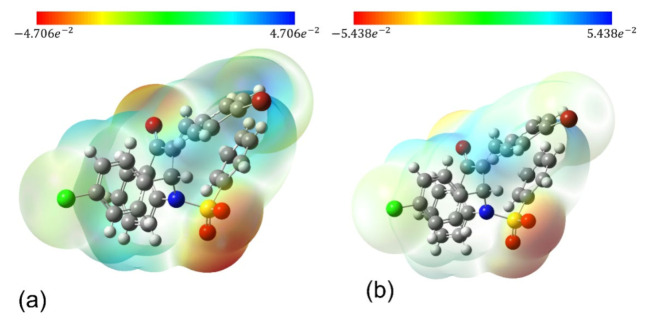
Molecular electrostatic potential (MEP) for QBCP isolated (**a**) and embedded (**b**) molecules.

**Figure 6 molecules-27-02379-f006:**
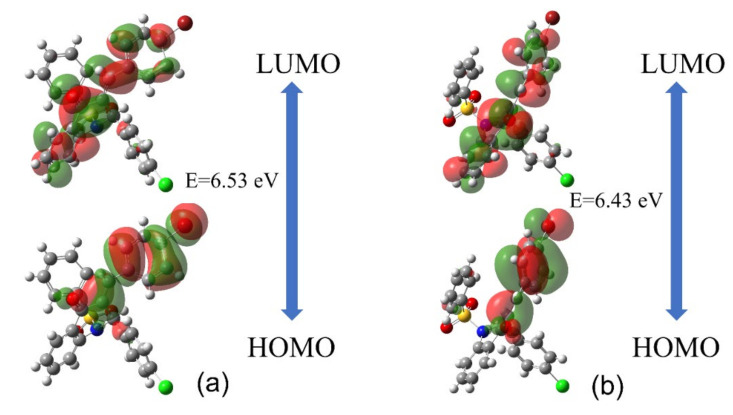
Plot of the HOMO and LUMO frontiers orbitals and the gap energies for the QBCP isolated (**a**) and embedded (**b**) molecules.

**Figure 7 molecules-27-02379-f007:**
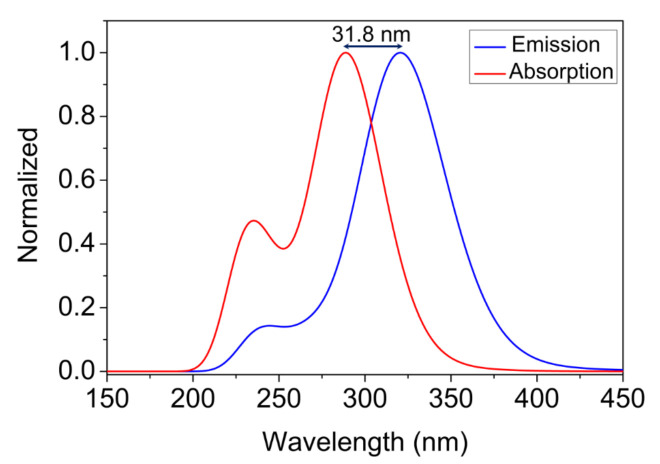
UV-Vis absorption and emission spectra of QBCP.

**Table 1 molecules-27-02379-t001:** Experimental details of QBCP.

Crystal Data	
Chemical formula	C_28_H_19_BrClNO_3_S
*M* _r_	564.86
Crystal system, space group	Monoclinic, *P*2_1_/*n*
Temperature (K)	296
*a*, *b*, *c* (Å)	11.7216 (4), 11.8413 (5), 18.7222 (8)
β (°)	102.166 (1)
*V* (Å^3^)	2540.26 (18)
*Z*	4
Radiation type	Mo *K*α
µ (mm^−1^)	1.84
Crystal size (mm)	0.40 × 0.40 × 0.35
Diffractometer	Bruker *APEX*-II CCD
No. of measured, independent and observed [*I* > 2σ(*I*)] reflections	70,693, 4473, 3747
*R* _int_	0.032
(sin θ/λ)_max_ (Å^−1^)	0.595
**Refinement**	
*R*[*F*^2^ > 2σ(*F*^2^)], *wR*(*F*^2^), *S*	0.036, 0.090, 1.03
No. of reflections	4473
No. of parameters	375
H-atom treatment	H atoms treated by a mixture of independent and constrained refinement
Δρ_max_, Δρ_min_ (e Å^−3^)	0.35, −0.36

**Table 2 molecules-27-02379-t002:** Selected dihedral angles for QBCP.

Dihedral Angle	Atoms	Value (°)	Conformation
*θ* _1_	C7—C8—C9—C11	175.9(5)	+Anti-Periplanar
*θ* _2_	C8—C9—C11—C12	147.3(7)	+Anti-Clinal
*θ* _3_	C7—C8—C10—C17	−77.7(2)	−Syn-Clinal
*θ* _4_	C8—C10—C17—C18	−17.5(3)	−Syn-Periplanar
*θ* _5_	C1—C2—N1—S1	−112.2(2)	−Anti-Clinal
*θ* _6_	C2—N1—S1—C23	72.6(5)	+Syn-Clinal
*θ* _7_	N1—S1—C23—C24	104(1)	+Anti-Clinal

**Table 3 molecules-27-02379-t003:** Geometric parameter of the C—H···O interactions in QBCP.

D—H···A	D—H (Å)	H···A (Å)	D···A (Å)	D—H···A (°)	Symmetry Code
C18—H18···O1	0.94(2)	2.51(2)	3.252(3)	137(2)	2−x,2-y,1−z
C6—H6···O3	0.94(2)	2.61(2)	3.396(3)	142(2)	1.5−x,−1/2 + y,1/2−z
C25—H25···O2	0.930	2.525	3.259(9)	136.0	2.5−x,1/2 + y,1.5−z

**Table 4 molecules-27-02379-t004:** Geometric parameters of the interactions involving π systems in QBCP.

X···Cg	Length (Å)	Symmetry Code
O3···Cg1	3.857	−1/2 + x,1.5−y,−1/2 + z
Br1···Cg2	3.861	1 + x,y,z
Cl1···Cg3	3.923	−1/2 + x,1.5−y,1/2 + z

**Table 5 molecules-27-02379-t005:** Charges (e) of selected groups of the QBCP’s isolated and embedded molecules.

Groups		Isolated	Embedded	Ratio Embedded/Isolated
1	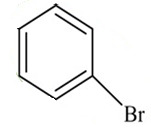	0.0786	0.1121	1.43
2	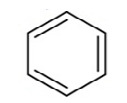	0.0073	0.0420	5.75
3	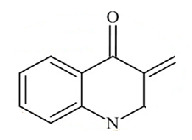	−0.0551	−0.0710	1.29
4	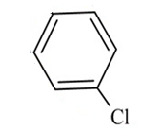	−0.0817	−0.0808	0.99
5	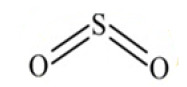	0.0474	−0.0024	−0.05

**Table 6 molecules-27-02379-t006:** The global reactivity descriptors. Energies are in eV units.

Descriptors	Isolated	Embedded
$\varepsilon_{HOMO}$	−8.27	−8.07
$\varepsilon_{LUMO}$	−1.73	−1.63
Ionization energy: $\left(I_{E}=-\varepsilon_{HOMO}\right)$	8.27	8.07
Electron affinity: $A_{E}=-\varepsilon_{LUMO}$	1.73	1.63
Global hardness: $\eta =\left(I_{E}-A_{E}\right)/2$	3.27	3.22
Chemical potential μ=($\left(I_{E}+A_{E}\right)/2)$	5.00	4.85
Global softness: σ=1/η	0.31	0.31
Global electrophilicity $\varpi =\mu^{2}/2\eta$	3.82	3.65
Electron donating capability $\left(\varpi^{-}\right)$	6.73	6.48
Electron accepting capability ($\varpi^{+})$	1.73	1.63

**Table 7 molecules-27-02379-t007:** Static and dynamic results for the refraction index (*n*), linear polarizability $\left(\alpha \;in\;10^{-24}esu\right),\;$ average second hyperpolarizabilities ($\gamma \;in\;10^{-36}esu)\;$ and third-order nonlinear susceptibility ($\chi^{3}\;in\;10^{-20}m^{2}/V^{2})$ of the QBCP-embedded molecule.

Electric Parameter	Static	*λ* (nm) 1064	*λ* (nm) 532
*n (* ω)	1.62	1.64	1.69
$\alpha \left(-\omega ;\omega \right)$	53.36	54.29	57.67
$\gamma \left(-\omega ;\omega ,0,0\right)$	67.06	76.24	127.11
$\gamma \left(-\omega ;\omega ,\omega ,-\omega \right)$	67.06	85.43	187.15
$\chi^{3}\left(-\omega ;\omega ,\omega ,-\omega \right)$	0.84	1.10	2.79

**Table 8 molecules-27-02379-t008:** The third-order nonlinear susceptibility $\left(\times 10^{-20}{\left(\frac{m}{V}\right)}^{2}\right)$ for several organic crystals at ω=0.086 a.u. (532 nm).

	$\chi^{\left(3\right)}\left(-\omega ;\omega ,\omega ,-\omega \right)$
QBCP (present work)	2.79
(2E)-3-(3-methylphenyl)-1-(4-nitrophenyl)prop-2-en-1-one [47]	2.77
(2E)-3-(3-methylphenyl)-1-(4-nitrophenyl)prop-2-en-1-one [46]	1.76
4,6-dichloro-2-(methylsulfonyl)pyrimidine [48]	0.57
(E)-3-(2-bromophenyl)-1-(2-((phenylsulfonyl)amine)-phenyl)prop-2-en-1-one [37]	0.26
1-(5-chlorothiophen-2-yl)-3-(2,3-dimethoxyphenyl)prop-2-en-1-one [47,49]	0.24
1-(5-chlorothiophen-2-yl)-3-(2,3-dichlorophenyl)prop-2-en-1-one [50]	0.16
2-(4-methylphenoxy)-N0-[(1E)-(4-nitrophenyl)methylene]acetohydrazide [51]	0.10
1-(4-aminophenyl)-3-(3,4,5-trimethoxyphenyl)prop-2-en-1-one [52]	0.09
(2E)-3 [4 (methylsulfanyl)phenyl]-1-(4-nitrophenyl)prop-2-en-1-one [53]	0.02
(2E)-1-(4-bromophenyl)-3-[4-methylsulfanyl) phenyl]prop-2-en-1-one [53]	0.02
(2E)-1-(3-bromophenyl)-3-[4 (methylsulfanyl) phenyl]prop-2-en-1-one [53]	0.02

## Data Availability

Not applicable.

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
