# Peer review of "Remarkable Nonlinear Properties of a Novel Quinolidone Derivative: Joint Synthesis and Molecular Modeling"

_molecules, 2022, doi:10.3390/molecules27082379_

Round 1
Reviewer 1 Report
The authors describe the synthesis and structure of a novel 4(1H) quinolinone derivative (QBCP). DFT was used to investigate the NLO properties of this compound.
Overall the article is well written, with a clear structure.
Regarding the Xray structure determination:
It is actually very uncommon to refine both the coordinates AND the B-factors of hydrogen atoms
I would be inclined to use Afix constraints for the position (for H not involved in H-bonds) and/or set the B factor to -1.2 or 1.5 as appropriate (or use a FVAR to refine an overall B-factor)
That said, it seems to refine to ok values, ...
What is however not correct is fixing the occupancy of C14 and Br1 (and C14A and Br1A) to 50% while refining the occupancy of the attached phenyl ring with a Free Variable. Now atoms with an occupancy of 0.565 or 0.435 are bonded to an atom at 0.5 occupancy!
Also all atoms of the bromophenyl ring should be given the same variable, now C11 and C11A are given FVAR 3 while the other atoms are given FVAR 4
This needs to be fixed and the corrected cif should be redeposited to the ccdc
2.4 The crystalline environment simulation
It is unclear whether the disorder was taken into account when simulating the crystalline environment, it should be explicitly mentioned.
Rather than calculating the effect on the different asymmetric units (for which no difference was to to be expected as they are symmetry related) it would be better to simulate the crystalline environment taking the different combinations of disorder into account, combining part 1/2 (bromophenyl) to part 3 /4 (phenyl) (combinations (1-3, 1-4 , 2-3, 2-4)
Figure 5 and by extension the paragraph on the Hirschfeld surface (p9 lines 237 - 246 ) can be removed as they do not add any additional info beyond what was already described in the section above (lines 208 - 236 + tables 3+4 and figures 3+4) . And in doing so section 2.3 can also be removed.
When describing the interactions involving the Pi-systems, consider the disordered parts as well, the C14A-Br1A ... Cg2 distance is actually shorter (3.6865(17)) than the one reported in table 4.
minor comments
p6 line 189 : all letters in space group names are to be written in italic
p7 line 197 ORTEP map -> ORTEP representation
Author Response
The authors describe the synthesis and structure of a novel 4(1H) quinolinone derivative (QBCP). DFT was used to investigate the NLO properties of this compound.
Overall the article is well written, with a clear structure.
Regarding the Xray structure determination:
It is actually very uncommon to refine both the coordinates AND the B-factors of hydrogen atoms I would be inclined to use Afix constraints for the position (for H not involved in H-bonds) and/or set the B factor to -1.2 or 1.5 as appropriate (or use a FVAR to refine an overall B-factor)
That said, it seems to refine to ok values, ...
Question 1: What is however not correct is fixing the occupancy of C14 and Br1 (and C14A and Br1A) to 50% while refining the occupancy of the attached phenyl ring with a Free Variable. Now atoms with an occupancy of 0.565 or 0.435 are bonded to an atom at 0.5 occupancy! Also all atoms of the bromophenyl ring should be given the same variable, now C11 and C11A are given FVAR 3 while the other atoms are given FVAR 4 This needs to be fixed and the corrected cif should be redeposited to the ccdc
Answer 1: Reviewed. We have refined the occupancy of C14 and Br1 (and C14A and Br1A) using the same Free Variable of the attached phenyl ring. Also, we undertaken the redeposit on CCDC.
Question 2: 2.4 The crystalline environment simulation. It is unclear whether the disorder was taken into account when simulating the crystalline environment, it should be explicitly mentioned.
Answer 2: Done as requested. We have included the following sentence in the manuscript: “It is worth to note that the QBCP geometry used in the SM method was retrieved from crystallographic file and possible disorder was not taken into account.”
Question 3: Rather than calculating the effect on the different asymmetric units (for which no difference was to to be expected as they are symmetry related) it would be better to simulate the crystalline environment taking the different combinations of disorder into account, combining part 1/2 (bromophenyl) to part 3 /4 (phenyl) (combinations (1-3, 1-4 , 2-3, 2-4)
Answer 3: Done as requested. In previous work we have showed that the results obtained taken into account the explicitly the unit cell or dimers do not significantly alter the nonlinear optical properties [1].
[1] Valverde, C.; Osório, F.A.P.; Fonseca, T.L.; Baseia, B. DFT Study of Third-Order Nonlinear Susceptibility of a Chalcone Crystal. Chemical Physics Letters 2018, 706, 170–174, doi:10.1016/j.cplett.2018.06.001.
Question 4: Figure 5 and by extension the paragraph on the Hirschfeld surface (p9 lines 237 - 246 ) can be removed as they do not add any additional info beyond what was already described in the section above (lines 208 - 236 + tables 3+4 and figures 3+4) . And in doing so section 2.3 can also be removed.
Answer 4: Reviewed. We have moved the Figure 5 to Support Information. Also, the section 2.3 (Hirschfeld surface) was condensed and merged as part of section 2.2 (Crystallographic characterization).
Question 5: When describing the interactions involving the Pi-systems, consider the disordered parts as well, the C14A-Br1A ... Cg2 distance is actually shorter (3.6865(17)) than the one reported in table 4.
Answer 5: Please, consider we selected only dominant disordered part on Table 4 in order to describe the interactions involving the Pi-systems.
Question 6: p6 line 189 : all letters in space group names are to be written in italic
Answer 6: Done as requested.
Question 7: p7 line 197 ORTEP map -> ORTEP representation
Answer 7: Done as requested.
Reviewer 2 Report
In their manuscript, the authors report the synthesis, single crystal X-ray diffraction, and computational analysis of linear and nonlinear optical properties of a novel quinolinone derivative. The experimental and computational methodologies are appropriate and the reported study does not show serious problems. However, the work does not differ substantially from several recent papers published by the authors, and appears to be a small incremental advance for a specific compound. The paper would be more exciting if comparisons between experimental and theoretical results would be reported for optical properties. This could be done, at least, for the absorption and emission of the isolated molecule, which are routine experiments.
Besides this, I have more specific comments on the manuscript:
- The quality of English should be polished throughout the text.
- Lines 100-101: From the sentence: “…the distance between a point on the surface and the nearest nucleus outside the surface, ??, or the nearest nucleus outside again the surface, ??…”, it is not clear what is the difference between ?? and ??. Please rephrase.
- Equations 8 and 9 should be presented on a single line.
- It should be explicitly mentioned in the text that optical properties are calculated using time-dependent DFT.
- Lines 180-181: “This basis set has shown to be accurate aiming at performing calculations of NLO and photophysical properties”: Is this remark holds for the basis set only (6-311++G(d,p)) or also for the chosen exchange-correlation functional? A reference is also missing here.
- Only the level of calculation used to calculate the optical properties is mentioned, not the one used to optimize the molecular geometries. Is it the same?
- Regarding the calculation of the emission spectrum: what is the emitting excited state?
- Were the molecular geometries of the ground and excited states validated against vibrational frequency calculations?
- Figure 9: I could not find any reason to present this figure using 3D histograms within a blur frame.
Author Response
In their manuscript, the authors report the synthesis, single crystal X-ray diffraction, and computational analysis of linear and nonlinear optical properties of a novel quinolinone derivative. The experimental and computational methodologies are appropriate and the reported study does not show serious problems. However, the work does not differ substantially from several recent papers published by the authors, and appears to be a small incremental advance for a specific compound. The paper would be more exciting if comparisons between experimental and theoretical results would be reported for optical properties. This could be done, at least, for the absorption and emission of the isolated molecule, which are routine experiments.
Besides this, I have more specific comments on the manuscript:
Question 1: The quality of English should be polished throughout the text.
Answer 1: The final version of the manuscript was revised by a native English speaker.
Question 2: Lines 100-101: From the sentence: “…the distance between a point on the surface and the nearest nucleus outside the surface, ??, or the nearest nucleus outside again the surface, ??…”, it is not clear what is the difference between ?? and ??. Please rephrase.
Answer 2: Done as requested.
Question 3: Equations 8 and 9 should be presented on a single line.
Answer 3: Done as requested.
Question 4: It should be explicitly mentioned in the text that optical properties are calculated using time-dependent DFT.
Answer 4: The phrase: ”Also, the UV-Vis absorption and emission spectra of QBCP molecule calculated at the time-dependent TD-DFT/CAM-B3LYP/6-311++G(d,p) level of theory in a vacuum are presented.” See section 2.3.
Question 5: Lines 180-181: “This basis set has shown to be accurate aiming at performing calculations of NLO and photophysical properties”: Is this remark holds for the basis set only (6-311++G(d,p)) or also for the chosen exchange-correlation functional? A reference is also missing here.
Answer 5: The CAM-B3LYP/6-311++G(d,p) level of theory has shown to be accurate aiming at performing calculations of NLO and photophysical properties [38,47].
Question 6: Only the level of calculation used to calculate the optical properties is mentioned, not the one used to optimize the molecular geometries. Is it the same?
Answer 6: All the computational calculations including the optical properties were performed at the CAM-B3LYP/6-311++G(d,p) level of theory as implemented at the Gaussian-16 program package.
Question 7: Regarding the calculation of the emission spectrum: what is the emitting excited state?
Answer 7: First excited state. The information was included in the manuscript (See section 2.3).
Question 8: Were the molecular geometries of the ground and excited states validated against vibrational frequency calculations?
Answer 8: Yes. The information was included in the manuscript (See section 2.3).
Question 9: Figure 9: I could not find any reason to present this figure using 3D histograms within a blur frame.
Answer 9: Reviewed. The figure has been removed and table 8 has been included
Round 2
Reviewer 2 Report
The authors improved their text and globally answered the questions I raised. In my opinion, the manuscript is now publishable.